# Bipolar Transurethral Enucleation of the Prostate: Is it a size-independent endoscopic treatment option for symptomatic benign prostatic hyperplasia?

Carolina Bebi[1], Matteo Turetti[1], Elena Lievore[1], Francesco Ripa[1], Lorenzo Rocchini[1], Matteo Giulio Spinelli[1], Elisa De Lorenzis[1,2], Giancarlo Albo[1,2], Fabrizio Longo[1], Franco Gadda[1], Paolo Guido Dell'Orto[1], Andrea Salonia[3], Emanuele Montanari[1,2], Luca Boeri[1] *

1 Department of Urology, Foundation IRCCS Ca' Granda–Ospedale Maggiore Policlinico, University of Milan, Milan, Italy, 2 Department of Clinical Sciences and Community Health, University of Milan, Milan, Italy, 3 Division of Experimental Oncology/Unit of Urology, URI, IRCCS Ospedale San Raffaele, Milan, Italy

* dr.lucaboeri@gmail.com

## Abstract

### Background

Bipolar Transurethral Enucleation of the Prostate (B-TUEP) is recommended as a first-choice treatment for benign prostatic obstruction in prostates >80 ml. Differently, B-TUEP is only considered as an alternative option after TURP for smaller prostates (30–80 ml). The aim of our study is to assess the relation between prostate size and surgical outcomes after B-TUEP.

### Methods

We performed a retrospective analysis of data collected from 172 patients submitted to B-TUEP. Patients were segregated according to tertiles of prostate volume (PV) ($\leq$60 ml, 61–110 ml, >110 ml). For each group we evaluated enucleation efficacy (enucleated weight/enucleation time), complication rates, urinary and sexual function parameters. Functional and sexual parameters were compared between groups at baseline, 1 and 3 months follow up. Descriptive statistics and linear and logistic regression models tested the association between PV and postoperative complications/outcomes.

### Results

Operative time and weight of enucleated adenomas increased along with prostate volumes (all p$\leq$0.01). Enucleation efficacy was higher in men with PV >110 ml compared to other groups (p$\leq$0.001). Length of hospital stay, catheterization time and rates of postoperative complications, such as transfusion and clot evacuation rates and bladder neck/urethral strictures, were comparable between groups. Urinary symptoms improved at 1-and 3-months in each group as compared to baseline evaluation (all p<0.01) but they did not differ according to PV. In each group maximum urinary flow and post-void residual volume significantly

**Data Availability Statement:** All relevant data are within the paper.

**Funding:** The authors received no specific funding for this work.

**Competing interests:** The authors have declared that no competing interests exist.

improved at 3 months compared to baseline (all p≤0.01), without differences according to PV. Sexual symptoms were similar between groups at each follow up assessment. At multivariable linear and logistic regression analysis, prostate volume was not associated with postoperative functional outcomes and complications. Conversely, patient's comorbid status and antiplatelet/anticoagulation use were independently associated with postoperative complications.

## Conclusion

According to our findings, B-TUEP should be considered a "size independent procedure" as it can provide symptom relief in men with prostates of all sizes with the same efficacy and safety profile.

## Introduction

Transurethral resection of the prostate (TURP) has been considered the gold standard surgical treatment of benign prostatic obstruction (BPO) for decades. This technique, however, is associated with cumulative short-term morbidity rates as high as 11.1% [1], which prompted clinical researchers to focus on equally effective, but safer alternatives. Among new surgical options, Bipolar Transurethral Enucleation of the Prostate (B-TUEP) has been proposed to exploit the advantages of bipolar electrocautery and the superiority of enucleation over resection [2]. Studies have shown its safety and efficacy in comparison to other forms of BPO surgery [3–6]. However, the majority of studies on B-TUEP focus on larger prostate sizes [7, 8]. As a matter of fact, according to the European Urological Association (EAU) guidelines treatment algorithms, B-TUEP is recommended, together with TURP and laser enucleation, as first choice in case of prostate volumes >80 ml, but it is only considered as an alternative option after TURP for smaller prostates (30–80 ml) [9]. To the best of our knowledge, there is a lack of studies comparing surgical outcomes after B-TUEP in patients with a wide range of prostate volume (from small to range volume). Nevertheless, the same principle has extensively been applied for enucleative techniques employing Holmium and Thulium laser energies [10–15] and laser enucleation is currently considered a size independent treatment option for BPH. Because prostate size at baseline has been shown to correlate with both perioperative and postoperative outcomes after BPO surgery, it is important to determine whether B-TUEP outcomes are also dependent on prostate volumes when we offer this technique as a treatment option for BPO relief.

The aim of our study is to comprehensively assess the relation between prostate size and surgical outcomes after B-TUEP in terms of complication rates, and modifications of urinary and sexual parameters on the basis of our 3 years single centre experience.

## Materials and methods

Between 01/11/2016 and 01/05/2019 a total of 172 consecutive white-European patients suffering from LUTS/BPO underwent B-TUEP in our institution. Clinical data, perioperative characteristics and surgical outcomes were prospectively collected for all patients and retrospectively analysed for the purpose of this study. For each subject we considered measured Body Mass Index (BMI), rates of preoperative urethral catheterization (POC), BPO-related drug use (alpha-blockers, 5-alpha reductase inhibitors or combination therapy) and significant comorbid conditions, which were scored with the Charlson Comorbidity Index (CCI;

categorized 0 vs. ≥1) [16]. The routine pre-operative assessment included measurement of prostate specific antigen (PSA), and evaluation of prostate volume (PV) and maximum urinary flow rate respectively by means of trans-rectal ultrasonography and uroflowmetry. Patients were also invited to complete the International Prostate Symptoms Score (IPSS) questionnaire in order to objectively quantify baseline LUTS severity [17]. The International Index of Erectile Function- Erectile Function (IIEF-EF) domain and the Male Sexual Health Questionnaire-Ejaculatory function (MSHQ-EJ) questionnaires were used to record erectile function and ejaculation characteristics [18, 19], whereas urinary incontinence was investigated by means of the International Consultation of Incontinence–Short Form questionnaire (ICIQ-SF) [20]. As recommended by current European Association of Urology Guidelines, we offered B-TUEP as a surgical option to relieve LUTS/BPO in men with prostate volumes >30 ml [9]. There was no predetermined upper limit on prostate size that could be treated by B-TUEP. Before surgery, urine culture was required for each patient and those positive to the test were treated on the basis of the antibiogram. All patients received a preoperative wide-spectrum antibiotic prophylaxis (second generation cephalosporin if not contraindicated).

All surgeries were performed in the same tertiary referral centre skin-to-skin by a single expert surgeon (P.D.) (>100 cases).

## Surgical technique

All surgeries were carried out with the Olympus UES-40 SurgMaster TUR system (Olympus Europa Holding GmbH, Hamburg, Germany). To perform enucleation, the surgeon employed either the standard tungsten wire loop or the B-TUEP loop, which consists of a spatula attached to the standard wire and is specifically designed to apply the required pressure to enucleate the adenoma and achieve haemostasis (Fig 1). The B-TUEP procedure was carried out

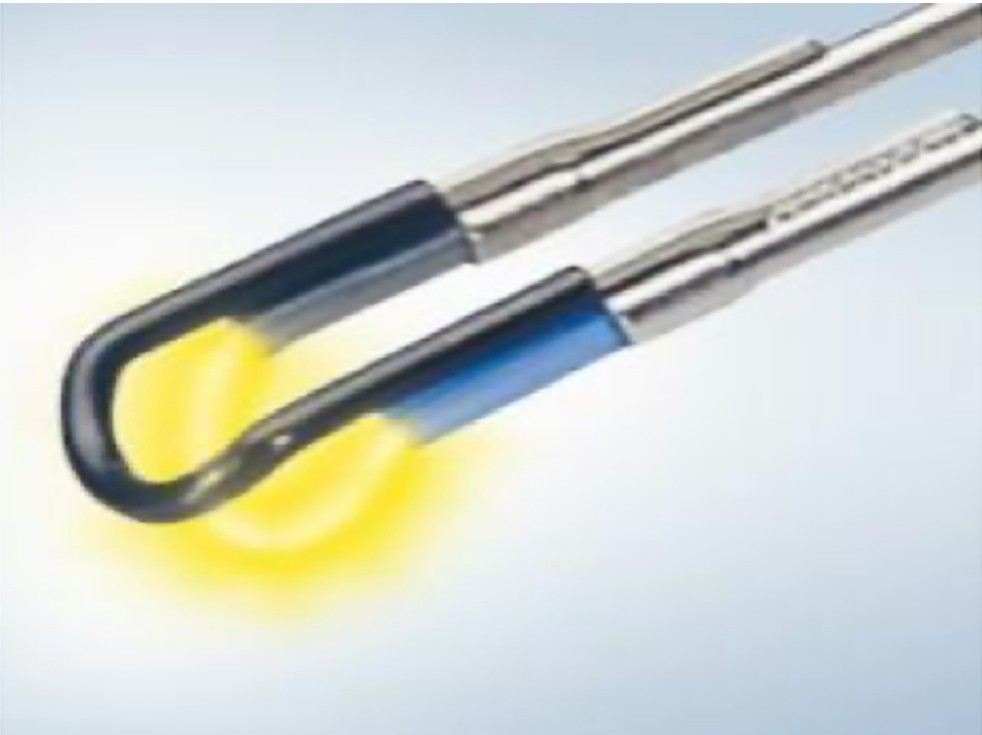

**Fig 1. The Bipolar Transurethral Enucleation of the Prostate (B-TUEP) loop.**

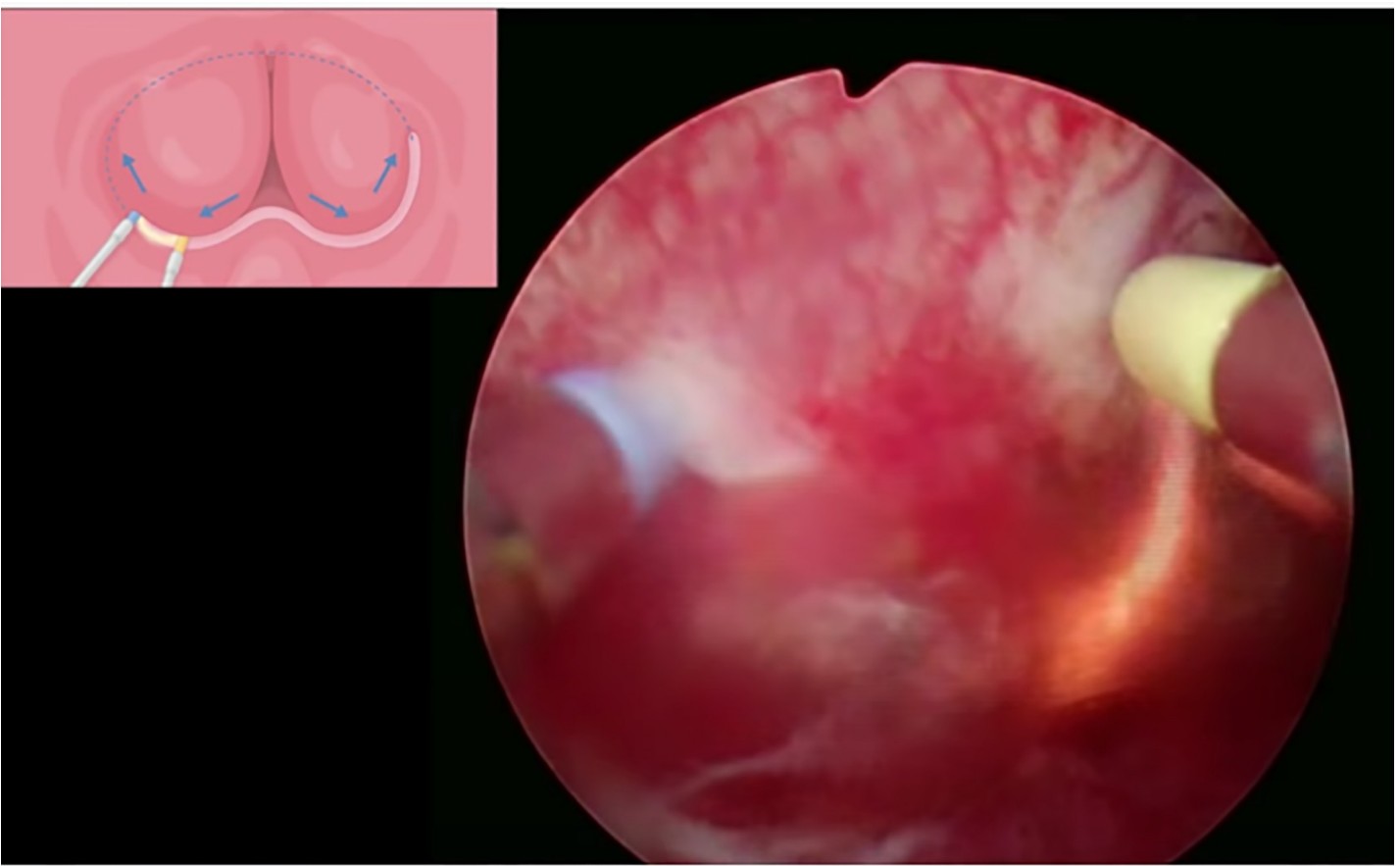

**Fig 2. After creating a groove at the 12 o'clock position, two additional grooves are performed at the 5 and 7 o'clock positions, laterally to the veru montanum.**

in the same fashion for all cases, regardless of prostate size, presence of prominent middle lobes or asymmetry. The first step of surgery is to create a groove at the 12 o'clock position, followed by two additional grooves at the 5 and 7 o'clock positions, laterally to the veru montanum (Fig 2). Next, the lateral lobes and the middle lobe, when present, are bluntly dissected circumferentially from the prostate apex towards the bladder following the plane of the capsule. This allows for the enucleation of the adenoma, which is gently torn-away by repeatedly pushing the loop against the adenoma with a circular motion (Fig 3). During this process, electrocautery is not in use, but it is solely applied for precise coagulation of crossing vessel that may bleed when the adenoma is separated from the rest of the prostate. After the process of enucleation is completed, the button electrode may be employed to limit bleeding, based on surgeon's preference.

After the tissue is released into the bladder, tissue morcellation is performed (Lumenis VersaCut Tissue Morcellator). After the procedure, a 22 Fr 3-way catheter is positioned for continuous bladder irrigation, which is continued overnight and weaned gradually as needed.

## Postoperative care

The indwelling catheter is most commonly removed on post-operative day 1 or 2 and patients are discharged after spontaneous voiding of urine. In case of significant gross haematuria, catheter removal was postponed on the basis of the treating physician's decision.

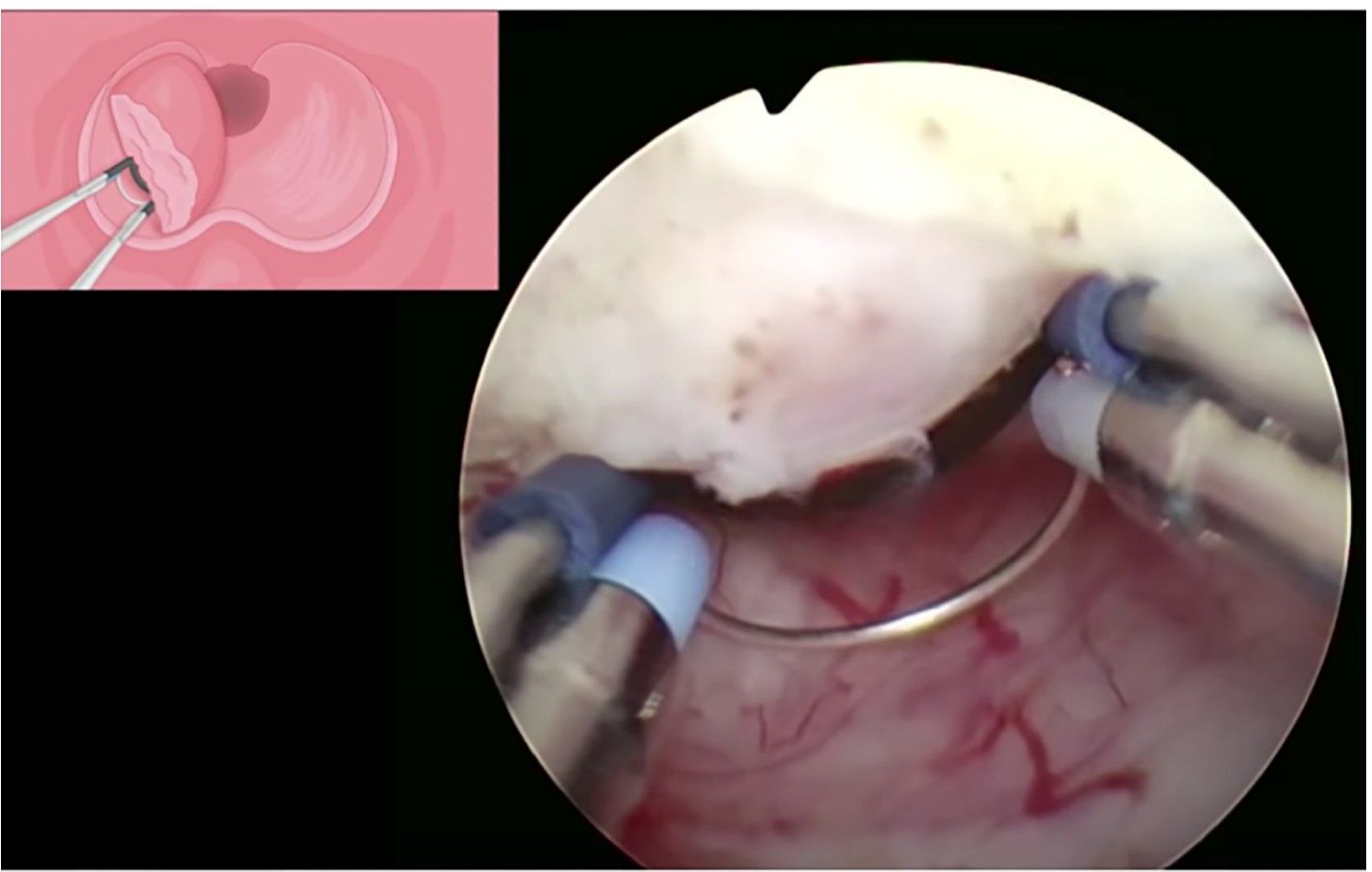

**Fig 3. Next, the lateral lobes of the prostate are bluntly dissected circumferentially from the prostate apex towards the bladder following the plane of the capsule.**

All patients were instructed to access the hospital emergency department in the event of post-surgical complications. Surgical complications were classified according to Dindo et al. [21]. As per standard clinical protocol, follow-up office/based visits were routinely scheduled 1, 3 and 12 months after surgery. Patients were asked to complete the psychometric question-naires at all follow-up assessments, whereas uroflowmetry and PSA were only repeated 3 and 12 months after surgery. The primary objective of the study was to investigate the relationship between prostate volume and surgical outcomes after B-TUEP. For the specific purpose of the study, patients were segregated according to tertiles of prostate volume (namely, ≤60 ml, 61–110 ml, and >110 ml). Perioperative outcomes included weight of the enucleated adenoma and calculation of the enucleation efficacy, expressed as enucleated weight/enucleation time. Among postoperative outcomes we recorded length of hospital stay, catheterization time and complication rates (i.e. incidence of transfusions, clot evacuation rates, development of stric-tures). Evaluated urinary parameters included the IPSS and ICIQ-SF scores, maximum urinary flow and post-void residual volume, whereas IIEF-EF scores were used to document changes in sexual parameters from baseline to follow-up.

Exclusion criteria were: patients older than 80 years old (N = 10); presence of a known pros-tate or bladder cancer (N = 1); neurogenic disorders or history of bladder disease or other uro-logic conditions likely to affect micturition (e.g. urethral stenosis, urinary incontinence, chronic bacterial prostatitis) (N = 8); concomitant antidepressant therapy (N = 3); and previ-ous surgical treatment for LUTS/BPO (N = 5).

Data were collected following the principles outlined in the Declaration of Helsinki. All patients signed an informed consent form agreeing to share their own anonymous information for future studies. The study was approved by the IRCCS Foundation Ca' Granda–Maggiore Policlinico Hospital Ethical Committee (Prot. 25508).

## Statistical analysis

Distribution of data was tested with the Shapiro-Wilk test. Data are presented as medians (interquartile range; IQR) or frequencies (proportions). Pre-, intra-, and post-operative variables were compared between the three groups (i.e. PV$\leq$ 60 ml, PV 61–110 ml and PV>110 ml) with the Fisher exact test and the Kruskal-Wallis test with multiple comparisons. In each group, potential differences in functional and sexual parameters at each follow-up assessment (baseline, 1 and 3 months) were evaluated with the paired t-test. Functional and sexual outcomes were compared between groups at each follow-up with the Kruskal-Wallis test. Spearman's correlation tested the association between clinical variables and prostate size. Univariable (UVA) and multivariable (MVA) linear regression analyses tested the associations between clinical predictors and 1-3-months post-surgery IPSS in the whole cohort. Similarly, UVA and MVA logistic regression analysis were used to identify potential predictors of post-operative complications (any). Statistical analyses were performed using SPSS v.26 (IBM Corp., Armonk, NY, USA). All tests were two sided and statistical significance level was determined at $p<0.05$.

## Results

Table 1 details preoperative characteristics of patients submitted to B-TUEP and categorized according to tertiles of prostate volume. Overall, 49 (28.4%), 74 (43.2%) and 49 (28.4%) men had a preoperative PV of $\leq$60 ml, 61–110 ml and >110 ml, respectively. Groups were comparable in terms of age, BMI, CCI and anticoagulation/antiplatelet (AC/AP) use. Serum PSA increased with increasing prostate size ($p\leq0.001$). A higher rate of men with PV >110 ml had POC as compared to those in the other groups (p = 0.01). The lowest preoperative maximum flow was reported in men with PV >110 ml ($p\leq0.001$). The distribution of BPH-related medications was similar among groups.

### Perioperative outcomes among B-TUEP patients according to prostate size

Operative time and the weight of the enucleated adenoma increased along with the categories of prostate volumes (all $p\leq0.01$) (Table 2). The enucleation efficacy was higher in men with PV >110 ml compared to those in the other groups ($p\leq0.001$). The percentage of PSA reduction was significantly correlated with enucleated adenoma (Spearman's Rho = 0.5, $p\leq0.001$). Length of hospital stay, catheterization time and rates of postoperative complications were comparable between groups (Table 2). In particular, transfusion and clot evacuation rates were independent of PV (S1 Table). The rate of late complications (bladder neck and urethral strictures) was similar among groups.

### Functional outcomes among B-TUEP patients

Preoperative total IPSS score was similar between groups (Table 3). After surgery, total IPSS score improved at 1-and 3-months in each group as compared to baseline evaluation (all p<0.01). However, at each follow up assessment, total IPSS score did not differ according to prostate volume. Both storage and voiding symptoms significantly improved after surgery, irrespective of PV (all p<0.01) (Table 3). The ICIQ-SF score was higher at 1 months after

**Table 1. Preoperative characteristics of the whole cohort of patients submitted to B-TUEP as segregated according to prostate volume (no. = 172).**

| | ≤ 60 ml | 61–110 ml | > 110 ml | p value* |
|---|---|---|---|---|
| No. of patients [No. (%)] | 49 (28.4) | 74 (43.2) | 49 (28.4) | |
| Age (years) | | | | 0.9 |
| Median (IQR) | 71 (65–74) | 71 (66–76) | 71 (64–76) | |
| Range | 55–80 | 51–83 | 51–85 | |
| BMI (kg/m2) | | | | 0.8 |
| Median (IQR) | 25.8 (23.8–27.8) | 25.0 (23.1–27.4) | 24.9 (23.8–27.8) | |
| Range | 20.3–39.6 | 17.6–38.1 | 19.3–35.0 | |
| CCI > = 1 [No. (%)] | 28 (57.1) | 42 (56.8) | 23 (46.9) | 0.5 |
| AP/AC therapy [No. (%)] | 21 (42.9) | 23 (31.1) | 13 (26.5) | 0.2 |
| Year of surgery [No. (%)] | | | | 0.7 |
| 2017 | 15 (30.6) | 29 (39.2) | 18 (36.7) | |
| 2018 | 16 (32.7) | 25 (33.8) | 20 (27.0) | |
| 2019 | 18 (36.7) | 20 (27.0) | 14 (28.6) | |
| PSA (ng/mL) | | | | ≤0.001 |
| Median (IQR) | 1.6 (0.8–3.6) | 3.8 (1.8–5.9) § | 4.6 (2.7–6.7) § | |
| Range | 0.2–12.5 | 0.1–15.3 | 0.9–21.5 | |
| POC [No. (%)] | 6 (12.2) | 22 (30.6) § | 18 (36.2) § | 0.01 |
| Duration of POC (months) | | | | 0.9 |
| Median (IQR) | 7.0 (5.0–8.0) | 7.0 (4.0–9.0) | 7.0 (4.0–13.0) | |
| Range | 0.0–12 | 3.0–16 | 6.0–20 | |
| BPH-related drugs [No. (%)] | | | | 0.8 |
| Alpha-blockers | 21 (44.4) | 27 (37.1) | 20 (40.9) | |
| 5-ARI | 6 (13.4) | 14 (18.5) | 4 (9.1) | |
| Combination | 22 (42.2) | 33 (44.3) | 25 (50.0) | |
| Prostate Volume (ml) | | | | ≤0.001 |
| Median (IQR) | 60 (50–60) | 80 (70–90) § | 130 (120–146) §, # | |
| Range | 30–60 | 65–110 | 115–260 | |
| Flow Max (mL/sec) | | | | ≤0.01 |
| Median (IQR) | 9.2 (7.9–13.0) | 7.3 (4.6–8.4) § | 5.1 (4.4–11.3) §,# | |
| Range | 3.5–20.0 | 2.9–11.9 | 1.9–20.2 | |
| Post-void residual volume (ml) | | | | 0.01 |
| Median (IQR) | 80 (30–100) | 100 (30–140) § | 115 (60–170) § | |
| Range | 0–800 | 0–900 | 0–800 | |
| Preoperative Hemoglobin (g/dL) | | | | 0.8 |
| Median (IQR) | 14.5 (13.8–15.5) | 14.5 (13.7–15.3) | 14.9 (13.6–15.5) | |
| Range | 11.3–18.1 | 10.9–17.1 | 10.0–16.6 | |

Keys: B-TUEP = bipolar transurethral enucleation of the prostate; BMI = body mass index; CCI = Charlson Comorbidity Index

AP/AG = Antiplatelet/Anticoagulation; PSA = Prostate specific antigen; POC = pre-operative catheterization

BPH = benign prostatic hyperplasia; 5-ARI = 5-alpha reductase inhibitors

*P value according to unpaired Kruskal Wallis test for continuous data and Fisher Exact test for categorical variables, as indicated.

§ p < 0.01 vs. ≤ 60 ml group

# p < 0.01 vs. 60–110 ml group

surgery in each group as compared to baseline (all p≤0.01), but it returned to preoperative value at 3 months. No differences were found in terms of ICIQ-SF scores at each follow up assessment according to prostate size.

**Table 2. Perioperative characteristics of the whole cohort of patients submitted to B-TUEP as segregated according to prostate volume (no. = 172).**

| | ≤ 60 ml (N = 49) | 61–110 ml (N = 74) | > 110 ml (N = 49) | p value* |
|---|---|---|---|---|
| Operative time (min) | | | | ≤ 0.01 |
| Median (IQR) | 85 (64–115) | 105 (90–126) § | 150 (123–167) §, # | |
| Range | 45–190 | 45–320 | 57–240 | |
| Enucleation time (min) | | | | ≤ 0.001 |
| Median (IQR) | 56 (40–71) | 70 (60–83) § | 90 (76–106) §, # | |
| Range | 20–120 | 30–213 | 40–140 | |
| Enucleated adenoma (ml) | | | | ≤ 0.001 |
| Median (IQR) | 30 (20–40) | 70 (45–90) § | 110 (90–150) §, # | |
| Range | 20–60 | 55–140 | 70–180 | |
| Enucleation efficacy (ml/min) | | | | ≤ 0.001 |
| Median (IQR) | 0.4 (0.3–0.7) | 0.6 (0.5–0.9) §, # | 0.8 (0.6–1.0) §, # | |
| Range | 0.1–1.1 | 0.2–1.5 | 0.4–2.1 | |
| Catheterization time (days) | | | | 0.1 |
| Median (IQR) | 2.0 (1.0–3.0) | 2.0 (1.0–3.0) | 2.0 (2.0–3.0) | |
| Range | 1.0–10.0 | 1.0–9.0 | 1–7 | |
| Length of stay (days) | | | | 0.6 |
| Median (IQR) | 3.0 (3.0–4.0) | 3.0 (3.0–4.0) | 4.0 (3.0–4.0) | |
| Range | 2.0–19.0 | 2.0–9.0 | 2–9 | |
| Hemoglobin drop (g/dL) | | | | 0.1 |
| Median (IQR) | 1.0 (0.5–1.9) | 1.0 (0.7–2.2) | 1.2 (0.9–2.3) | |
| Range | 0.0–5.5 | 0.0–4.8 | 0.0–5.5 | |
| PSA reduction from baseline (%) | | | | 0.001 |
| Median (IQR) | 38.0 (20–65) | 60.0 (45–81) § | 84 (54–90) §, # | |
| Range | 10.0–93.0 | 11.0–97.0 | 15.0–98.1 | |
| Overall Complications [No. (%)] | 10 (20.4) | 12 (16.2) | 10 (20) | 0.8 |
| Complication severity [No. (%)] | | | | 0.5 |
| Clavien Dindo I | 1 (2.0) | 5 (6.8) | 2 (4.1) | |
| Clavien Dindo II | 3 (6.1) | 5 (6.8) | 5 (10.2) | |
| Clavien Dindo IIIa | 6 (12.2) | 3 (4.1) | 3 (6.1) | |
| Transfusion rate [No. (%)] | 1 (2.0) | 0 (0.0) | 1 (2.0) | 0.6 |

Keys: B-TUEP = bipolar transurethral enucleation of the prostate; PSA = Prostate specific antigen *P value according to unpaired Kruskal Wallis test for continuous data and Fisher Exact test for categorical variables, as indicated.

§ p < 0.01 vs. ≤ 60 ml group

# p < 0.01 vs. 60–110 ml group

In each group maximum urinary flow and post-void residual volume significantly improved at 3 months compared to baseline (all p≤0.01), without differences according to PV.

Preoperative IIEF-EF scores were similar between groups. After surgery, IIEF-EF scores were comparable to baseline values at each follow-up assessment irrespective of the study group (Table 3).

## Clinical predictors of postoperative IPSS and complications

Table 4 reports UVA and MVA liner regression analyses showing the associations between study variables and 1-3-months post-surgery IPSS in the whole cohort. Prostate volume was not associated with postoperative IPSS in men submitted to B-TUEP. This was also the case for patient's age, CCI, rates of POC and AC/AP use and preoperative IPSS scores.

**Table 3. Functional and sexual characteristics of the whole cohort of patients as segregated according to prostate size [median (IQR)].**

| | | ≤ 60 ml | | 61–110 ml | | > 110 ml | p value* |
|---|---|---|---|---|---|---|---|
| **Functional outcomes** | | | | | | | |
| Total IPSS score | | | | | | | |
| Preoperative | n = 49 | 18.0 (15–27) | n = 74 | 18.0 (14–24) | n = 49 | 20.0 (10–30) | 0.8 |
| 1 month | n = 48 | 6.0 (3–15) § | n = 74 | 7.0 (2–10) § | n = 49 | 8.0 (4–15) § | 0.4 |
| 3 months | n = 48 | 6.0 (3–13) § | n = 72 | 5.0 (3–10) § | n = 48 | 7.0 (5–16) § | 0.2 |
| IPSS-storage score | | | | | | | |
| Preoperative | | 8.0 (6–9) | | 7.0 (5–11) | | 7.0 (4–11) | 0.8 |
| 1 month | | 5.0 (2–8) § | | 5.0 (4–8) § | | 5.0 (3–9) § | 0.7 |
| 3 months | | 4.0 (3–7) § | | 4.0 (2–7) § | | 4.0 (3–11) § | 0.4 |
| IPSS-voiding score | | | | | | | |
| Preoperative | | 10.0 (6–12) | | 10.0 (5–12) | | 8.0 (5–14) | 0.9 |
| 1 month | | 4.0 (1–8) § | | 1.0 (0–4) § | | 2.0 (1–6) § | 0.3 |
| 3 months | | 3.0 (1–6) § | | 1.0 (0–2) § | | 2.0 (0–4) § | 0.2 |
| ICIQ-SF | | | | | | | |
| Preoperative | | 1.0 (0–7) | | 0.0 (0–4) | | 0.0 (0–5) | 0.4 |
| 1 month | | 3.0 (0–9) § | | 2.0 (0–7) § | | 3.0 (0–9) § | 0.3 |
| 3 months | | 0.0 (0–6) # | | 0.0 (0–5) # | | 0.0 (0–5) # | 0.6 |
| Flow Max (mL/sec) | | | | | | | |
| 3 months | n = 48 | 24.0 (14–35) § | n = 71 | 25.0 (11–30) § | n = 48 | 23.0 (10–29) § | 0.1 |
| Post-void residual volume (ml) | | | | | | | |
| 3 months | | 0.0 (0–30) § | | 0.0 (0–23) § | | 0.0 (0–26) § | 0.7 |
| **Sexual outcomes** | | | | | | | |
| IIEF-EF score | | | | | | | |
| Preoperative | | 23.0 (5–29) | | 23.0 (5–28) | | 22.0 (5–28) | 0.5 |
| 1 month | | 20.0 (4–28) | | 20.0 (6–29) | | 21.0 (4–28) | 0.7 |
| 3 months | | 20.0 (5–29) | | 19.0 (5–27) | | 20.0 (4–27) | 0.7 |

Keys: IPSS = International Prostatic Symptoms Score; ICIQ-SF = The International Consultation of Incontinence–Short Form

IIEF-EF = International Index of Erectile Function- Erectile Function domain

*P value according to unpaired Kruskal Wallis test

§ p < 0.01 vs. baseline. P value according to paired t-test

**Table 4. Univariable and multivariable linear regression models (beta; p value [95%CI]) predicting IPSS at 1 and 3 months after surgery.**

| | 1-month IPSS | | 3-months IPSS | |
|---|---|---|---|---|
| | UVA model | MVA model | UVA model | MVA model |
| Age | 0.21; 0.89 [-0.24–0.21] | 0.12; 0.37 [-0.14–0.38] | 0.14; 0.22 [-0.10–0.37] | 0.11; 0.49 [-0.21–0.43] |
| CCI ≥1 | -0.92; 0.51 [-3.74–1.89] | | -0.96; 0.53 [-4.06–2.13] | |
| POC | 0.03; 0.98 [-3.73–3.47] | -2.61; 0.26 [-7.29–2.07] | -1.34; 0.45 [-4.93–2.25] | -3.13; 0.22 [-8.33–2.06] |
| Prostate size | | | | |
| ≤ 60 ml | Ref | Ref | Ref | Ref |
| 61–110 ml | 0.31; 0.38 [-1.54–1.91] | 0.26; 0.22 [-2.97–1.41] | 0.82; 0.31 [-1.37–1.73] | 0.51; 0.27 [-1.09–2.06] |
| > 110 ml | 0.65; 0.73 [-2.19–2.49] | 0.69; 0.42 [-1.55–2.93] | 0.87; 0.14 [-1.98–3.73] | 0.53; 0.55 [-1.71–2.79] |
| AP/AC | 1.34; 0.36 [-1.61–4.29] | | 1.93; 0.24 [-1.33–5.19] | |
| Preoperative IPSS | -0.18; 0.11 [-0.41–0.05] | -0.14; 0.23 [-0.36–0.09] | -0.17; 0.21 [-0.44–0.11] | -0.11; 0.51 [-0.43–0.22] |

Keys: UVA = Univariate model; MVA = Multivariate model, CCI = Charlson Comorbidity Index

AP/AG = Antiplatelet/Anticoagulation; POC = pre-operative catheterization; IPSS = International Prostatic Symptoms Score.

**Table 5. Univariable and multivariable logistic regression models predicting postoperative complications (any) after surgery.**

| | Odds Ratio | 95% CI | p value | Odds Ratio | 95% CI | p value |
|---|---|---|---|---|---|---|
| | | UVA | | | MVA | |
| Age | 1.03 | 0.97; 1.11 | 0.21 | 1.02 | 0.95; 1.08 | 0.59 |
| CCI $\geq$1 | 3.78 | 1.53; 9.31 | <0.01 | 2.78 | 1.05; 7.45 | 0.04 |
| Operative time | 1.01 | 0.99; 1.01 | 0.67 | 1.01 | 0.99; 1.02 | 0.15 |
| Prostate size | | | | | | |
| $\leq$ 60 ml | Ref | | | Ref | | |
| 61–110 ml | 0.75 | 0.69; 1.91 | 0.53 | 0.66 | 0.52; 1.99 | 0.46 |
| > 110 ml | 0.81 | 0.58; 2.18 | 0.9 | 0.76 | 0.61; 2.91 | 0.69 |
| AP/AC | 2.83 | 1.29; 6.21 | <0.01 | 2.69 | 1.08; 6.76 | 0.03 |

Keys: UVA = Univariate model; MVA = Multivariate model, CCI = Charlson Comorbidity Index

AP/AG = Antiplatelet/Anticoagulation

On the contrary, multivariable analysis revealed that CCI$\geq$1 (OR 2.78; p = 0.04) and AC/AP use (OR 2.69; p = 0.03) were independently associated with postoperative complications, after accounting for age, operative time and prostate volume (Table 5).

## Discussion

In this study we found that B-TUEP is an excellent surgical option for BPO treatment. According to our findings, this technique can be considered as a "size independent procedure" since it is able to provide symptom relief in men with prostates of all sizes with the same efficacy and safety profile. Transurethral enucleation of the prostate with bipolar energy is well known to be effective in larger prostate sizes [4, 3, 6, 7], whereas only a few studies documented its efficacy in smaller prostates [22, 23].

Our study, however, was motivated by the complete lack of studies that aimed to directly compare outcomes of B-TUEP according to prostate volume.

In terms of complication rates, our findings are comparable to results reported for other enucleative techniques [8, 23]. Rates of perioperative complications, such as transfusion and clot evacuation were low, as only 2 patients in the whole cohort required transfusions and 4 patients required clot evacuation. The incidence of late complications such as bladder neck and urethral strictures was also limited, 4.1% in the $\leq$60 ml group, 4% in the 61–110 ml group, 4.1% in the >110ml group). As already reported, B-TUEP is safe in patients under AC/AP therapy [24], however, current results reported that AC/AP use and patients with comorbid conditions (as depicted by the CCI score) were at higher risk of postoperative complications, yet irrespective of prostate size. Notably, according to our analysis, complication rates, length of hospital stay, and catheterization time after surgery were comparable in all groups. While for trans-urethral resection of the prostate rates of complications have long been known to increase with higher prostate volumes [1, 25], more recent findings on enucleative techniques found no direct relation between complication rates and prostate volumes [10, 13, 26], which is consistent with our findings.

Additionally, in our cohort, urinary parameters such as IPSS score, maximum urinary flow and post-void residual volume also appeared to improve independently form prostate size.

Erectile function, as expressed by changes in the IIEF-EF scores, did not change after surgery, which is consistent with previously reported findings for most studies on BPO surgery [27–30].

One of the most prominent aspects of our findings is that the enucleation efficacy increased along with prostate volume categories. This parameter is expressed as a simple fraction

consisting of a numerator (enucleated weight) and a denominator (enucleation time) and this ratio was found to increase in larger prostates. If the enucleation is performed correctly, the increase in enucleated weight, i.e. the numerator, is expected to be proportional to the increase in volume. Interestingly, according to our findings, when prostates become larger, the corresponding increase in enucleation time, i.e. the denominator, must be smaller in proportion to the increase in volume. As a consequence, we may deduce that smaller adenomas take more time to be enucleated if compared to bigger ones. In other words, smaller prostates appear to be more difficult to enucleate. This concept corroborates a notion that has already been reported in previous studies [31], and is also often reported by surgeons as a mere intra-operative perception. A study by Hirasawa et al. [23] demonstrated that enucleation efficacy not only increased as prostate size increased, but also improved markedly when the surgeons experience level exceeded 50 cases. This implies that technical proficiency is paramount in order to perform B-TUEP.

Of note, as often recommended [32], surgeons tend to tackle larger prostates after having experience with smaller glands, which may represent a confounding factor for the measurement of operative time, due to the impact of the surgeon's learning curve on velocity. However, in our cohort, the number of small, intermediate and large prostates was evenly distributed throughout the years, therefore enucleation time could not be influenced by the surgeon's experience.

The clinical implications of our study are several. First, we conducted the first thorough investigation of functional and sexual outcomes following B-TUEP performed in men categorized according to different prostate volume. Indeed, our findings showed that B-TUEP is a size independent procedure. Second, we showed, for the first time, that enucleation efficacy during B-TUEP was higher in larger prostates as compared to smaller ones. These findings corroborate the difficulties of enucleating small adenomas that surgeons experience in the everyday clinical practice, thus potentially suggesting that the beginning of the learning curve of B-TUEP should be focused on large prostates.

Our study is not devoid of limitations. First, it was designed as a retrospective, non-randomised study, with the intrinsic limitations of its nature. Likewise, clinical homogeneity of the population might have influenced our results. Moreover, it describes the experience of a single surgeon in a single centre, therefore larger studies across different centres are needed in order to confirm our findings.

## Conclusions

B-TUEP is a "size independent procedure" since it can be performed in prostates of all volumes with comparable safety profile and functional results. The enucleation efficacy is higher for larger prostates with no effect on surgical outcomes. Prostate volume was not associated with postoperative functional outcomes and complication rates, while CCI≥1 and AC/AP therapy emerged as the only independent predictors of complications after B-TUEP. Future studies should also stratify their results on the basis of prostate size in order to determine if and how changes in volume may affect BPO surgery and whether prostate volume represents a parameter to be taken into account for adequate patient selection.

## Supporting information

**S1 Table. Descriptive characteristics of postoperative complications according to prostate size (N = 172).**
(DOCX)

## Author Contributions

**Conceptualization:** Carolina Bebi, Luca Boeri.

**Data curation:** Carolina Bebi, Matteo Turetti, Elena Lievore, Francesco Ripa, Lorenzo Roc-chini, Matteo Giulio Spinelli, Elisa De Lorenzis, Giancarlo Albo, Fabrizio Longo, Franco Gadda, Paolo Guido Dell'Orto, Andrea Salonia, Emanuele Montanari, Luca Boeri.

**Methodology:** Andrea Salonia, Emanuele Montanari, Luca Boeri.

**Supervision:** Carolina Bebi, Andrea Salonia, Emanuele Montanari, Luca Boeri.

**Writing – original draft:** Carolina Bebi.

**Writing – review & editing:** Carolina Bebi.

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
