## [Decision Letter · Decision Letter 0]

9 Apr 2021

PONE-D-21-05323

Bipolar Transurethral Enucleation of the Prostate: is it a size-independent endoscopic treatment option for symptomatic benign prostatic hyperplasia?

PLOS ONE

Dear Dr. Boeri,

Thank you for submitting your manuscript to PLOS ONE. After careful consideration, we feel that it has merit but does not fully meet PLOS ONE’s publication criteria as it currently stands. Therefore, we invite you to submit a revised version of the manuscript that addresses the points raised during the review process.

We look forward to receiving your revised manuscript.

Kind regards,

Henry Woo

Academic Editor

PLOS ONE

Journal Requirements:

2.  Please provide the full name of the institution that participants were recruited from."

3. Thank you for providing the date(s) when patient medical information was initially recorded. Please also include the date(s) on which your research team accessed the databases/records to obtain the retrospective data used in your study

Reviewers' comments:

Reviewer's Responses to Questions

**Comments to the Author**

1. Is the manuscript technically sound, and do the data support the conclusions?

Reviewer #1: Yes

Reviewer #2: Yes

2. Has the statistical analysis been performed appropriately and rigorously? 

Reviewer #1: Yes

Reviewer #2: Yes

3. Have the authors made all data underlying the findings in their manuscript fully available?

Reviewer #1: Yes

Reviewer #2: Yes

4. Is the manuscript presented in an intelligible fashion and written in standard English?

Reviewer #1: Yes

Reviewer #2: Yes

5. Review Comments to the Author

Reviewer #1: This study aimed to assess the relation between prostate size and surgical outcomes after Bipolar Transurethral Enucleation of the Prostate (B-TUEP). According to the conclusions of the authors, prostate volume was not associated with postoperative functional outcomes and complication rates. Thus, the current study may extend the current knowledge regarding the surgical outcomes of B-TUEP based on baseline prostatic volumes. However, there are still some major problems which need to be addressed.

1. In the Abstract, there is no description about comparison of efficacies of B-TUEP according to the baseline prostatic volumes. Please, add it in the Result section of the Abstract.

2. The introduction section of this manuscript needs to be clearer and more focused. Thus, the introduction section of this manuscript needs to be improved.

3. Also, the Discussion section needs to be further improved and summarized, focusing on the major findings of the study.

Reviewer #2: Bipolar Transurethral Enucleation of the Prostate: is it a size-independent endoscopic treatment option for symptomatic benign prostatic hyperplasia?

Abstract:

Retrospective study of 172 patients undergoing bipolar transurethral enucleation of the prostate (B-TUEP), aim of the study was to assess relationship between prostatic size and surgical outcomes following this technique.

Should change classification from quartiles to tertiles give that there’s only 3 groups (<60, 61-110, >110)

Outcomes of interest – efficiency (weight/time), complication rates, urinary/sexual function, LOS

Introduction:

I’d specify which laser energies (holmium, thulium etc) have been previously demonstrated as size-independent and I’d include Humpreys et al - Holmium Laser Enucleation of the Prostate—Outcomes Independent of Prostate Size? (2008) as another study for size independence.

Materials and Methods:

Small spelling error – ultrasonography and uroflowmetry (line 86)

Utilization of standardized metrics is helpful for cross study comparisons

Would specify which antibiotic was commonly used

Surgical Technique:

Intra-operative images or illustrations would be very helpful

I’d include a new header for postoperative care and data analysis to separate from the surgical technique section.

Again groups are better described as tertiles

Results:

Table 1 could be presented more cleanly if not across 2 pages (same for Table 2)

Discussion

Agree that part of the difficulty with small gland enucleation is probably represents a more challenging operation from an efficacy standpoint.

Clinical homogeneity of the patient population should be included as a potential weakness in the discussion.

6. PLOS authors have the option to publish the peer review history of their article (what does this mean?). If published, this will include your full peer review and any attached files.

Reviewer #1: No

Reviewer #2: **Yes: **Ryan Dobbs

---

## [Author Response · Author response to Decision Letter 0]

10 Apr 2021

Dr Emily Chenette

Deputy Editor-in-Chief, PLOS ONE

Dr Henry Woo

Academic Editor

PLOS ONE

Milan, April 10th, 2021

Dear Dr Emily Chenette,

Dear Dr Henry Woo,

please find enclosed the revised version of the manuscript titled “Bipolar Transurethral Enucleation of the Prostate: is it a size-independent endoscopic treatment option for symptomatic benign prostatic hyperplasia?” (PONE-D-21-05323- Authors: Carolina Bebi, et al) to be considered for publication in PLOS ONE.

We are very grateful to the Reviewers for their insightful comments to our paper.

List of the changes made in the manuscript:

REVIEWER #1

COMMENT#1. 

In the Abstract, there is no description about comparison of efficacies of B-TUEP according to the baseline prostatic volumes. Please, add it in the Result section of the Abstract.

A1. We thank the Reviewer#1 for this important comment. We have revised the Abstract accordingly.

COMMENT#2. The introduction section of this manuscript needs to be clearer and more focused. Thus, the introduction section of this manuscript needs to be improved

A2. We thank the Reviewer#1 for this comment. We have revised the Introduction section of the manuscript accordingly.

COMMENT#3. Also, the Discussion section needs to be further improved and summarized, focusing on the major findings of the study

A3. We thank the Reviewer#1 for this comment. We have revised the Discussion section of the manuscript accordingly.

REVIEWER #2

COMMENT#1. 

Should change classification from quartiles to tertiles give that there’s only 3 groups (<60, 61-110, >110)

A1. We thank the Reviewer#2 for this important comment. We have revised the text accordingly.

COMMENT#2. 

I’d specify which laser energies (holmium, thulium etc) have been previously demonstrated as size-independent and I’d include Humpreys et al - Holmium Laser Enucleation of the Prostate—Outcomes Independent of Prostate Size? (2008) as another study for size independence.

A2. We thank the Reviewer#2 for this important comment. The Introduction section of the manuscript has been revised accordingly. 

COMMENT#3. 

Small spelling error – ultrasonography and uroflowmetry (line 86)

Utilization of standardized metrics is helpful for cross study comparisons

Would specify which antibiotic was commonly used

A3. We thank the Reviewer#2 for this comment. We have revised the Methods section as suggested.

COMMENT#4. 

Surgical Technique:

Intra-operative images or illustrations would be very helpful

A4. We thank the Reviewer#2 for this comment. We have included Figures 1 to 3 in the new version of the manuscript

COMMENT#5. 

I’d include a new header for postoperative care and data analysis to separate from the surgical technique section.

Again groups are better described as tertiles

A5. We thank the Reviewer#2 for this comment. We have revised the Methods section as suggested.

COMMENT#6. 

Table 1 could be presented more cleanly if not across 2 pages (same for Table 2)

A6. We thank the Reviewer#2 for this comment. We have revised the Results section as suggested.

COMMENT#7. 

Agree that part of the difficulty with small gland enucleation is probably represents a more challenging operation from an efficacy standpoint.

Clinical homogeneity of the patient population should be included as a potential weakness in the discussion.

A7. We thank the Reviewer#2 for this comment. We have revised the Discussion section as suggested.

We hope that the paper is now suitable to be considered for publication in the Original Articles section of PLOS ONE.

Sincerely yours,

Luca Boeri on behalf of all the authors

Luca Boeri, M.D.,

IRCCS Foundation Ca’ Granda, Ospedale Maggiore Policlinico, Department of Urology

University of Milan

Via della Commenda 15, 20122 Milan, Italy

Tel. +39 02 55034501; Fax +39 02 50320584

Email: dr.lucaboeri@gmail.com

---

## [Editor Report · Decision Letter 1]

28 May 2021

Bipolar Transurethral Enucleation of the Prostate: is it a size-independent endoscopic treatment option for symptomatic benign prostatic hyperplasia?

PONE-D-21-05323R1

Dear Dr. Boeri,

We’re pleased to inform you that your manuscript has been judged scientifically suitable for publication and will be formally accepted for publication once it meets all outstanding technical requirements.

Kind regards,

Henry Woo

Academic Editor

PLOS ONE
---

## [Editor Report · Acceptance letter]

1 Jun 2021

PONE-D-21-05323R1 

Bipolar Transurethral Enucleation of the Prostate: is it a size-independent endoscopic treatment option for symptomatic benign prostatic hyperplasia? 

Dear Dr. Boeri:

I'm pleased to inform you that your manuscript has been deemed suitable for publication in PLOS ONE. Congratulations! Your manuscript is now with our production department. 

Kind regards, 

on behalf of

Dr. Henry Woo 

Academic Editor

PLOS ONE